# Increased Energy Efficiency of a Backward-Feed Multiple-Effect Evaporator Compared with a Forward-Feed Multiple-Effect Evaporator in the Cogeneration System of a Sugar Factory

**Somchart Chantasiriwan**

Department of Mechanical Engineering, Thammasat University, Pathum Thani 12121, Thailand; somchart@engr.tu.ac.th

**Abstract:** The cogeneration system of a sugar factory consists of boiler, steam turbine, and sugar juice evaporation process. The multiple-effect evaporator used for the conventional sugar juice evaporation process is the forward-feed multiple-effect evaporator, in which steam and sugar juice flow in the same direction. The main objective of this paper is to investigate the energy efficiency of the backward-feed multiple-effect evaporator, in which steam and sugar juice flow in opposite directions, compared with that of the forward-feed multiple-effect evaporator. Mathematical models are developed for both multiple-effect evaporators, and used to compare the performances of two cogeneration systems that use the forward-feed and backward-feed multiple-effect evaporators. The forward-feed multiple-effect evaporator requires extracted steam from a turbine at one pressure, whereas the backward-feed multiple-effect evaporator requires steam extraction at two pressures. Both evaporators have the same total heating surface area, process the same amount of sugar juice, and operate at the optimum conditions. It is shown that the cogeneration system that uses the backward-feed multiple-effect is more energy efficient than the cogeneration system that uses the forward-feed multiple-effect because it yields more power output for the same fuel consumption.

**Keywords:** heat exchanger; mathematical model; energy efficiency; counter-current

## 1. Introduction

The sugar juice evaporation process in raw sugar manufacturing removes a substantial amount of water from diluted juice using thermal energy from steam condensation. The inputs of this process are diluted juice at the ambient temperature and saturated steam at a high pressure. The outputs of this process are raw sugar, molasses, saturated vapor at low pressure, and condensate. This process is a component of a cogeneration system, which also consists of boiler and steam turbine. Bagasse, which is a by-product of sugar juice extraction process, is used as the main fuel for the boiler. High-pressure steam generated by the boiler is expanded in the extraction-condensing steam turbine. Steam extracted from the turbine at a specified pressure is sent to the evaporation process, and the remaining steam is condensed at a low pressure in the condenser. The primary outputs of the cogeneration system are raw sugar from the evaporation process and power output from the steam turbine. The energy efficiency of the cogeneration system can be measured by these outputs relative to the amount of bagasse required by the boiler.

The evaporation process takes place in multiple-effect evaporator, in which evaporation occurs in many effects. Multiple-effect evaporators in the sugar industry operate in the forward-feed arrangement, in which steam and sugar juice flow in the same direction. Advantages of this flow arrangement include the exposure of the most concentrated juice to the lowest saturated water vapor temperature in the last

effect, juice flashing on entering the second and subsequent effects, which assists flow circulation, and favorable pressure profile that does not require pumping between effects [1]. There have been several investigations on the improvement of energy efficiency of the cogeneration system by reducing steam consumption in the evaporator. Urbaniec et al. [2] recommended retrofitting the evaporation process to improve heat recovery. Higa et al. [3] performed an analysis showing that steam consumption could be reduced by increasing the number of effects. Mechanical vapor compression and thermal vapor compression have also been suggested by, respectively, Palacios-Bereche et al. [4] and Chen and Ruan [5] as methods of increasing the energy efficiency of a multiple-effect evaporator. Steam consumption by a multiple-effect evaporator can also be minimized by distributing heating surface areas optimally in the process. Previous investigations by Chantasiriwan [6–9] have demonstrated the existence of the optimum distribution of heating surface areas that results in the minimum steam consumption.

A multiple-effect evaporator can also operate in the backward-feed arrangement, in which steam and sugar juice flow in opposite directions. It can be shown that if the hot and cold fluids in a counter-flow heat exchanger and a parallel-flow heat exchanger have identical inlet temperatures, the log-mean temperature difference is larger in the counter-flow heat exchanger. Since heat transfer between the hot and cold fluids is proportional to the log-mean temperature difference, the counter-flow heat exchanger can transfer more heat than the parallel-flow heat exchanger that has the same heating surface area. As a result, if the inlet temperatures of the hot and cold fluids are fixed, the counter-flow heat exchanger will yield lower outlet temperature of the hot fluid and higher outlet of the cold fluid. Superior energy efficiency of the counter-current flow or backward-feed arrangement in the multiple-effect evaporator was demonstrated by Bhargawa et al. [10]. The backward-feed arrangement is used in several industries [10–13].

In this paper, performances of cogeneration systems that use the backward-feed multiple-effect evaporator and the forward-feed multiple-effect evaporator in their evaporation processes are compared. Mathematical models of both multiple-effect evaporators are developed for this purpose. Both cogeneration systems consume the same amount of bagasse, and produce the same amount of raw sugar. Therefore, the cogeneration system that produces higher power output has higher energy efficiency. It will be demonstrated by simulation that the backward-feed multiple-effect evaporator is responsible for the higher energy efficiency of the cogeneration system.

## 2. Forward-Feed Multiple-Effect Evaporator

Figure 1 shows the multiple-effect evaporator operating in the conventional forward-feed or co-current flow arrangement, which is also known as the forward-feed multiple-effect evaporator. Saturated steam and vapor (denoted by solid line) flow from the first effect (E1) to the last effect (E4), and so does sugar juice (denoted by dashed line). The evaporator requires a supply of steam at pressure $p_0$, which is extracted from an extraction-condensing steam turbine. Although extracted steam is usually superheated, it is assumed that the steam has been de-superheated, and is saturated at the inlet of the evaporator. In each effect, steam or vapor condensation at pressure $p_i$ causes the evaporation of water in sugar juice at a lower pressure $p_{i+1}$. Vapor leaving E1 is sent to the pan stage (P), the secondary juice heater (H2), and the second effect (E2), Vapor leaving E2 is sent to the primary heater (H1) and the third effect (E3). All vapor leaving E3 is sent to the final effect (E4). Syrup leaving E4 is pumped to a syrup tank before being sent to P.

Condensate (denoted by a dotted line) occurs after steam or vapor condensation. Condensate from E1 is used as feed water for the boiler. Condensate from P, H1, and E2 is sent to F1, whereas condensate from F1, H2, and E3 is sent to F2. Each flash tank receives condensate at pressure $p_i$ to produce vapor and condensate at a lower pressure $p_{i+1}$. Condensate from F2 and E4 is sent to a storage tank.

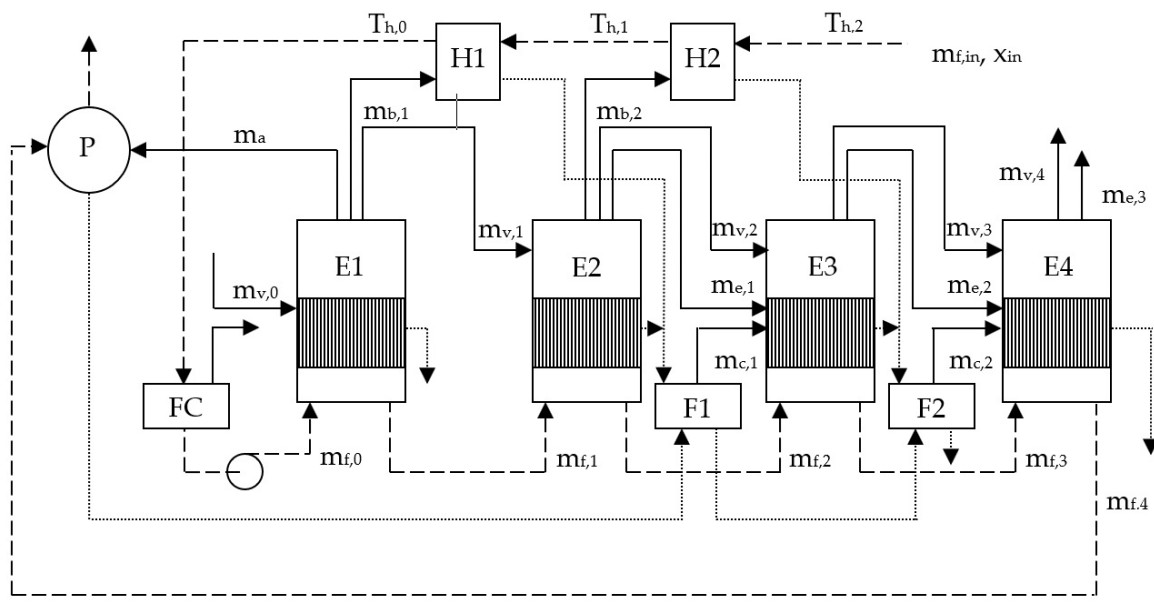

**Figure 1.** Forward-feed multiple-effect evaporator.

The evaporator requires juice heating in H1 and H2, which raises juice temperature from the ambient temperature ($T_{h,2} = T_{amb}$) to the saturation temperature ($T_{h,0} = 103\,°C$) corresponding to 112.7 kPa. Juice pressure is then decreased to the atmospheric pressure (101.3 kPa) after passing through FC. The model of the juice heaters is identical with the previous model presented by Chantasiriwan [9], and is represented by the following equations:

$$m_{b,1} h_{vl}(p_1) = m_{f,in} c_{p,f}\left(T_{h,0} - T_{h,1}\right),\tag{1}$$

$$m_{b,2} h_{vl}(p_2) = m_{f,in} c_{p,f}\left(T_{h,1} - T_{h,2}\right),\tag{2}$$

$$U_{h,1} A_{h,1} \frac{\left(T_{h,0} - T_{h,1}\right)}{\ln\left[\left(T_1 - T_{h,1}\right)/\left(T_1 - T_{h,0}\right)\right]} = m_{f,in} c_{p,f}\left(T_{h,0} - T_{h,1}\right),\tag{3}$$

$$U_{h,2} A_{h,2} \frac{\left(T_{h,1} - T_{h,2}\right)}{\ln\left[\left(T_2 - T_{h,2}\right)/\left(T_2 - T_{h,1}\right)\right]} = m_{f,in} c_{p,f}\left(T_{h,1} - T_{h,2}\right).\tag{4}$$

The correlation for $U_{h,i}$ given by Hugot [14]:

$$U_{h,i} = 0.007 T_i \left(\frac{u}{1.8}\right)^{0.8},\tag{5}$$

with the juice velocity ($u$) equal to 2.0 m/s is used in this model. After leaving H1, the juice pressure ($p_{in}$) is a little above the atmospheric pressure ($p_{out}$). The juice is allowed to flash in FC, resulting in a reduced mass flow rate ($m_{f,0}$) that is determined from

$$m_{f,0} = m_{f,in}\left[1 - f(p_{in}, p_{out})\right],\tag{6}$$

where

$$f(p_i, p_j) = \left[\frac{h_l(p_i) - h_l(p_j)}{h_v(p_j) - h_l(p_j)}\right].\tag{7}$$

As a consequence, the juice concentration at the inlet to the first effect ($x_0$) is related to the juice concentration at the inlet to the primary juice heater ($x_{in}$) as follows:

$$x_i m_{f,i} = m_{f,in} x_{in}, \tag{8}$$

where $i$ = 0–4.

Before entering E1, juice pressure is increased from $p_{out}$ to $p_1$ by a pump. It is assumed that pumping does not increase juice temperature. Therefore, juice temperature is $T_{out}$ at the inlet to E1. It should be noted that $T_{out}$ is lower than the saturation temperature ($T_1$) in E1. Therefore, part of the heating surface area in E1 is used for juice heating, which increases juice temperature to the saturation temperature before water evaporation can occur.

The energy balance equation for E1 is

$$(1 - \varepsilon)(m_{v,0} - m_{x,0})h_{vl}(p_0) + m_{f,0}\left(h_{f,1}^{(in)} - h_{f,1}^{(out)}\right) = \left(m_a + m_{v,1} + m_{b,1}\right)\left[h_v(p_1) - h_{f,1}^{(out)}\right]. \tag{9}$$

The heat-loss coefficient ($\varepsilon$) is assumed to be 0.015. The specific enthalpy of sugar juice ($h_{f,i}$) is the product of the average specific heat capacity of sugar juice and juice temperature in effect $i$, which is assumed to be saturation temperature. Saturated juice temperature is larger than the temperature of saturated liquid water at the same pressure due to the concentration of dissolved solids in juice and the hydrostatic pressure. It is expressed as

$$T_f = -227.03 + \frac{3816.44}{18.3036 - \ln[7.5(p + \rho g H/2000)]} + \frac{2x}{100 - x}, \tag{10}$$

where $x$ is concentration of dissolved solids in sugar juice, $g$ is gravitational acceleration, $\rho$ is juice density, and $H$ is liquid level in evaporator vessels, which is assumed to be 0.3 m. The variable $m_{x,0}$ in Equation (9) accounts for the mass flow rate of steam required to raise juice temperature from $T_{out}$ to $T_{f,1}^{(in)}$. It is determined from

$$m_{x,0} = \frac{m_{f,0} c_{p,f}\left(T_{f,1}^{(in)} - T_{out}\right)}{(1 - \varepsilon)h_{vl}(p_0)}. \tag{11}$$

The energy balance equation for E2 is

$$(1 - \varepsilon)m_{v,1}h_{vl}(p_1) + m_{f,1}\left(h_{f,2}^{(in)} - h_{f,2}^{(out)}\right) = \left(m_{v,2} + m_{b,2}\right)\left[h_v(p_2) - h_{f,2}^{(out)}\right]. \tag{12}$$

Evaporation in E2 is driven by saturated vapor from E1. Unlike E1, all of the heating surface area in E2 is used for water evaporation because incoming juice pressure ($p_1$) is larger than the juice pressure ($p_2$) in E2. Consequently, flash evaporation causes juice to be at the saturation temperature upon entering each of these effects. It also results in the reduced mass flow rate of juice in E2, which is

$$m_{f,1} = \left(m_{f,0} - m_a - m_{v,1} - m_{b,1}\right)[1 - f(p_1, p_2)]. \tag{13}$$

Flash evaporation also results in saturated vapor at pressure $p_2$, which goes to E3. Its mass flow rate ($m_{e,1}$) is

$$m_{e,1} = m_{f,0} - m_{f,1} - m_{v,1} - m_{b,1} - m_a. \tag{14}$$

The energy balance equation for E3 is

$$(1 - \varepsilon)(m_{v,2} + m_{e,1} + m_{c,1})h_{vl}(p_2) - h_{f,2}^{(out)} + m_{f,2}\left(h_{f,3}^{(in)} - h_{f,3}^{(out)}\right) = m_{v,3}\left[h_v(p_3) - h_{f,3}^{(out)}\right]. \tag{15}$$

Like E2, there is flash evaporation in E3, which results in the following reduced mass flow rate of juice in E3,

$$m_{f,2} = \left(m_{f,1} - m_{v,2} - m_{b,2}\right)[1 - f(p_2, p_3)], \tag{16}$$

and the following mass flow rate ($m_{e,2}$) of saturated vapor at pressure $p_3$ that goes to E4,

$$m_{e,2} = m_{f,1} - m_{f,2} - m_{v,2} - m_{b,2}. \tag{17}$$

Mass balance and energy balances of F1 yield the following expression for the vapor mass flow rate from F1,

$$m_{c,1} = \left(m_{v,1} + m_{b,1} + m_a\right)f(p_1, p_2). \tag{18}$$

Similarly, the energy balance equation for E4 is

$$(1 - \varepsilon)(m_{v,3} + m_{e,2} + m_{c,2})h_{vl}(p_3) + m_{f,3}\left(h_{f,4}^{(in)} - h_{f,4}^{(out)}\right) = m_{v,4}\left[h_v(p_4) - h_{f,4}^{(out)}\right], \tag{19}$$

where

$$m_{f,3} = \left(m_{f,2} - m_{v,3}\right)[1 - f(p_3, p_4)], \tag{20}$$

$$m_{c,2} = \left(m_{f,0} - m_{f,1} + m_{v,2} + m_{b,2}\right)f(p_2, p_3). \tag{21}$$

The mass flow rate of concentrated sugar juice or syrup leaving E4 is

$$m_{f,4} = m_{f,3} - m_{v,4}. \tag{22}$$

The surface area of E1 that is used for juice heating is $A_{x,0}$. It is determined by using the same model of juice heaters in Equations (3) and (4):

$$U_{h,0}A_{x,0}\frac{(T_1 - T_{out})}{\ln[(T_0 - T_{out})/(T_0 - T_1)]} = m_{f,0}c_{p,f}(T_1 - T_{out}). \tag{23}$$

Consequently, the heat transfer equation in E1 becomes

$$U_1(A_1 - A_{x,0})\left(T_0 - T_{f,1}^{(out)}\right) = (1 - \varepsilon)(m_{v,0} - m_{x,0})h_{vl}(p_0). \tag{24}$$

The correlation of $U_i$ of Robert evaporator is given by Wright [15]:

$$U_i = 0.000049(110 - x_i)^{1.1616}\left(T_{f,i}^{(out)}\right)^{1.0808}\left(T_{i-1} - T_{f,i}^{(out)}\right)^{0.266}. \tag{25}$$

Since there is no juice heating in E2, E3, and E4, the heat transfer equations in these effects are

$$U_2A_2\left(T_1 - T_{f,2}^{(out)}\right) = (1 - \varepsilon)m_{v,1}h_{vl}(p_1), \tag{26}$$

$$U_3A_3\left(T_2 - T_{f,3}^{(out)}\right) = (1 - \varepsilon)(m_{v,2} + m_{e,1} + m_{c,1})h_{vl}(p_2), \tag{27}$$

$$U_4A_4\left(T_3 - T_{f,4}^{(out)}\right) = (1 - \varepsilon)(m_{v,3} + m_{e,2} + m_{c,2})h_{vl}(p_3). \tag{28}$$

Most of the water content of the syrup coming from E4 is removed in the pan stage, yielding raw sugar and molasses. The pan stage is modeled as a single-effect evaporator. Vapor bled from the first effect is used to evaporate water in the syrup so that the mass fraction of dry substance is increased to around 91%. In practice, crystallization in P is usually carried out in three stages [1]. In each stage, water is added, and heat loss occurs. Therefore, the ideal amount of vapor bleeding required for P must

be multiplied by a correction factor to obtain the actual amount of vapor bleeding. If the correction factor is 2,

$$m_a = \frac{2m_{f,4}(1 - x_4/91)h_{vl}(p_4)}{h_{vl}(p_1)}. \tag{29}$$

The first-effect pressure ($p_1$) is normally required to equal or exceed the minimum value in order to create a sufficient temperature difference that provides a driving force in the crystallization process. It is assumed in this paper that this value is 150 kPa.

## 3. Backward-Feed Multiple-Effect Evaporator

Figure 2 shows the multiple-effect evaporator operating in the backward-feed or counter-current flow arrangement, which is also known as the backward-feed multiple-effect evaporator. Steam and vapor flow from E1 to E4, whereas sugar juice flows from E4 to E1. Condensate from E1 is used as feed water for the boiler. Condensate from E2 is sent to F1, whereas condensate from F1 and E3 is sent to F2. Condensate from F2 and E4 is sent to a storage tank. No vapor bleeding is required for juice heating and crystallization in this evaporator. Therefore, vapor leaving each effect except the last one is used only for evaporation in the next effect. The evaporator requires extracted steam at pressure $p_0$ for E1 and extracted steam at a specified pressure ($p_a$) for P. This means that the backward-feed multiple-effect evaporator requires extracted steam at two pressures. By contrast, the forward-feed multiple-effect evaporator requires extracted steam at only one pressure.

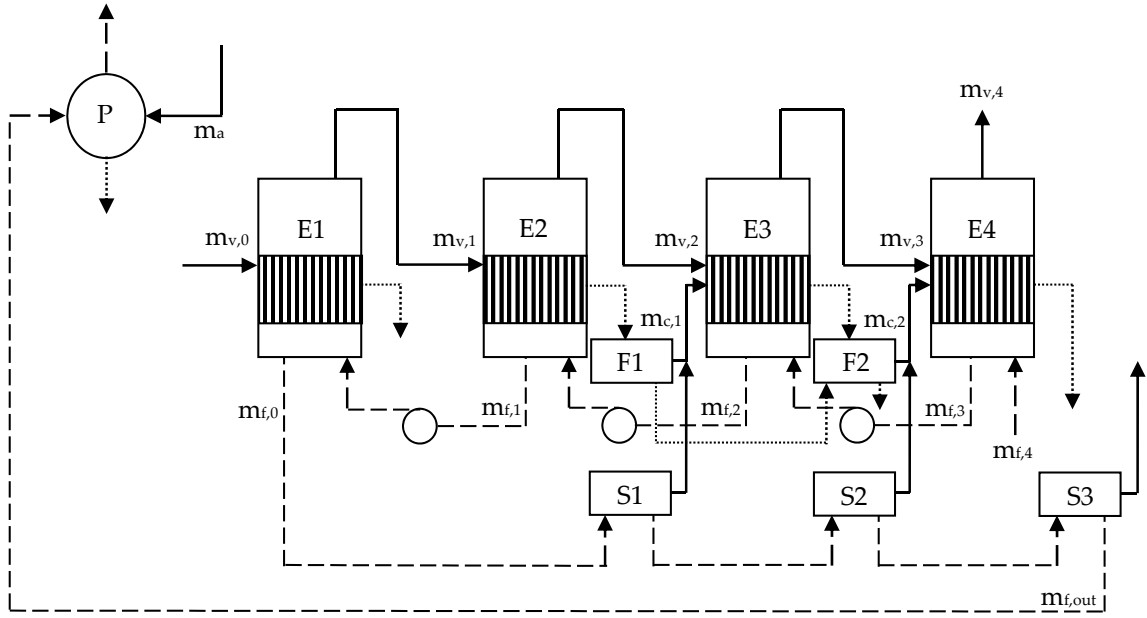

**Figure 2.** Backward-feed multiple-effect evaporator.

The juice temperature entering effect *i* is $T_{x,i}$. The juice temperature entering E4 is assumed to be the ambient temperature,

$$T_{x,4} = T_{amb}. \tag{30}$$

Since this temperature is lower than the saturation temperature corresponding to the pressure in E4, part of the heating surface area in E4 is used for juice heating that increases juice temperature to the saturation temperature before water evaporation can occur. Juice leaving each effect is pumped to the next effect. Juice temperature is assumed to be unaltered after being pumped. Therefore,

$$T_{x,i} = T_{f,i+1}^{(out)}, \tag{31}$$

for $i = 2$, 3, and 4. Since juice temperature at the inlet to each effect is lower than the saturation temperature in that effect, juice heating also occurs in E1, E2, and E3.

The system shown in Figure 2 makes use of solution flash, which has been demonstrated by Ruan et al. [12] to improve the system performance. Saturated juice leaving E1 at pressure $p_1$ is sent to S1, which reduces its pressure to $p_2$, and produces saturated vapor at the same pressure that is supplied to E3. Saturated juice leaving S1 and S2 are sent, respectively, to S2 and S3, which reduce juice pressure to $p_3$ and $p_4$ before the resulting concentrated juice or syrup is sent to a syrup tank. Saturated vapor leaving S2 at pressure $p_3$ is supplied to E4, whereas saturated vapor leaving S3 pressure $p_4$ is unused.

The mathematical model of the backward-feed multiple-effect evaporator differs from that of the forward-feed multiple-effect evaporator in equations of mass and energy balances, which are

$$m_{f,1} = m_{f,0} + m_{v,1}, \tag{32}$$

$$m_{f,2} = m_{f,1} + m_{v,2}, \tag{33}$$

$$m_{f,3} = m_{f,2} + m_{v,3}, \tag{34}$$

$$m_{f,4} = m_{f,3} + m_{v,4}, \tag{35}$$

$$(1-\varepsilon)(m_{v,0} - m_{x,1})h_{vl}(p_0) + m_{f,1}\left(h_{f,1}^{(in)} - h_{f,1}^{(out)}\right) = m_{v,1}\left[h_v(p_1) - h_{f,1}^{(out)}\right], \tag{36}$$

$$(1-\varepsilon)(m_{v,1} - m_{x,2})h_{vl}(p_1) + m_{f,2}\left(h_{f,2}^{(in)} - h_{f,2}^{(out)}\right) = m_{v,2}\left[h_v(p_2) - h_{f,2}^{(out)}\right], \tag{37}$$

$$(1-\varepsilon)(m_{v,2} + m_{c,1} - m_{x,3})h_{vl}(p_2) + m_{f,3}\left(h_{f,3}^{(in)} - h_{f,3}^{(out)}\right) = m_{v,3}\left[h_v(p_3) - h_{f,3}^{(out)}\right], \tag{38}$$

$$(1-\varepsilon)(m_{v,3} + m_{c,2} - m_{x,4})h_{vl}(p_3) + m_{f,4}\left(h_{f,4}^{(in)} - h_{f,4}^{(out)}\right) = m_{v,4}\left[h_v(p_4) - h_{f,4}^{(out)}\right]. \tag{39}$$

The mass flow rates of saturated vapor from F1 and F2 are

$$m_{c,1} = \left(m_{v,1} + m_{f,0}\right)f(p_1, p_2), \tag{40}$$

$$m_{c,2} = \left(m_{v,1} + m_{v,2} + m_{f,0}\right)f(p_2, p_3). \tag{41}$$

The mass flow rate of steam or vapor ($m_{x,i}$) required to raise juice temperature from a sub-cooled value to the saturation temperature in effect $i$ is determined from

$$m_{x,i} = \frac{m_{f,i}c_{p,f}\left(T_{f,i}^{(in)} - T_{x,i}\right)}{(1-\varepsilon)h_{vl}(p_{i-1})}. \tag{42}$$

After accounting for heating surface areas that are required to raise juice temperatures to saturation temperatures in all effects, the heat transfer equations in E1, E2, E3, and E4 become

$$U_1(A_1 - A_{x,1})\left(T_0 - T_{f,1}^{(out)}\right) = (1-\varepsilon)(m_{v,0} - m_{x,1})h_{vl}(p_0), \tag{43}$$

$$U_2(A_2 - A_{x,2})\left(T_1 - T_{f,2}^{(out)}\right) = (1-\varepsilon)(m_{v,1} - m_{x,2})h_{vl}(p_1), \tag{44}$$

$$U_3(A_3 - A_{x,3})\left(T_2 - T_{f,3}^{(out)}\right) = (1-\varepsilon)(m_{v,2} + m_{c,1} - m_{x,3})h_{vl}(p_2), \tag{45}$$

$$U_4(A_4 - A_{x,4})\left(T_3 - T_{f,4}^{(out)}\right) = (1-\varepsilon)(m_{v,3} + m_{c,2} - m_{x,4})h_{vl}(p_3). \tag{46}$$

It is assumed the correlation of the overall heat transfer coefficient for Robert evaporator given by Wright [15] is valid in the backward-feed multiple-effect evaporator. However, $x_i$ must be replaced by

$x_{i-1}$ in Equation (25) because $x_{i-1}$ is the juice concentration at the outlet of effect $i$ in the backward-feed multiple-effect evaporator. The surface area in effect i that is required to raise juice temperature from $T_{x,i}$ to $T_{f,i}^{(in)}$ is determined by using the same model of juice heaters in Equations (3) and (4),

$$U_{h,i-1}A_{x,i}\frac{\left(T_{f,i}^{(in)} - T_{x,i}\right)}{\ln\left[(T_{i-1} - T_{x,i})/\left(T_{i-1} - T_{f,i}^{(in)}\right)\right]} = m_{f,i}c_{p,f}\left(T_{f,i}^{(in)} - T_{x,i}\right). \tag{47}$$

After flowing past S1, S2, and S3, the mass flow rate of juice decreases from $m_{f,0}$ to

$$m_{f,out} = m_{f,0}[1 - f(p_1, p_2)][1 - f(p_2, p_3)][1 - f(p_3, p_4)], \tag{48}$$

and juice concentration increases from $x_0$ to

$$x_{out} = \frac{x_0}{[1 - f(p_1, p_2)][1 - f(p_2, p_3)][1 - f(p_3, p_4)]}. \tag{49}$$

Thermal energy required for evaporation in the pan stage of the backward-feed multiple-effect evaporator must be supplied by extracted steam at the specified pressure ($p_a$) from the turbine because there is no vapor bleeding from the evaporator. The model of the pan stage of the backward-feed multiple-effect evaporator is the same as that of the forward-feed multiple-effect evaporator. Therefore,

$$m_a = \frac{2m_{f,out}(1 - x_{out}/91)h_{vl}(p_4)}{h_{vl}(p_a)}. \tag{50}$$

The extracted steam pressure should be at least equal to the design pressure of the pan stage. It is assumed that this pressure is 150 kPa.

## 4. Cogeneration Systems

In comparing different forward-feed multiple-effect evaporators, in which the pressure of extracted steam supplied to the first effect of the evaporator and the pressure of vapor leaving the last effect are the same, it is sufficient to compare the steam economy, defined as the ratio of the amount of evaporated water to the amount of extracted steam. In that case, the evaporator that has the largest steam economy is the most energy efficient. However, steam economy is not suitable performance parameter for comparing the backward-feed multiple-effect evaporator and the forward-feed multiple-effect evaporator because both evaporators may require extracted steam at different pressures. An appropriate performance parameter is that of a cogeneration system, in which a multiple-effect evaporator is a component.

The schematics of cogeneration systems that use the forward-feed and backward-feed multiple-effect evaporators, to be referred to as the FF system and BF system, are shown in Figure 3. Both systems produce high-pressure steam using bagasse boilers (B). Superheated steam leaving the boilers at mass flow rate $m_s$, pressure $p_s$, and temperature $T_s$ is expanded in extraction-condensing steam turbines (T). Steam extracted at mass flow rate $m_{v,0}$ and pressure $p_0$ is supplied to the first effect of multiple-effect evaporator (MEE) of the FF system. The mass flow rate of remaining steam ($m_c$) that is sent to condenser (C) is $m_s - m_{v,0}$. The BF system requires extracted steam at two pressures. Extracted steam at mass flow rate $m_{v,0}$ and pressure $p_0$ is sent to the first effect of multiple-effect evaporator. Extracted steam at mass flow rate $m_a$ and pressure $p_a$ is sent to the pan stage. Therefore, the mass flow rate of the remaining steam ($m_c$) that is sent to the condenser in the BF system is $m_s - m_{v,0} - m_a$.

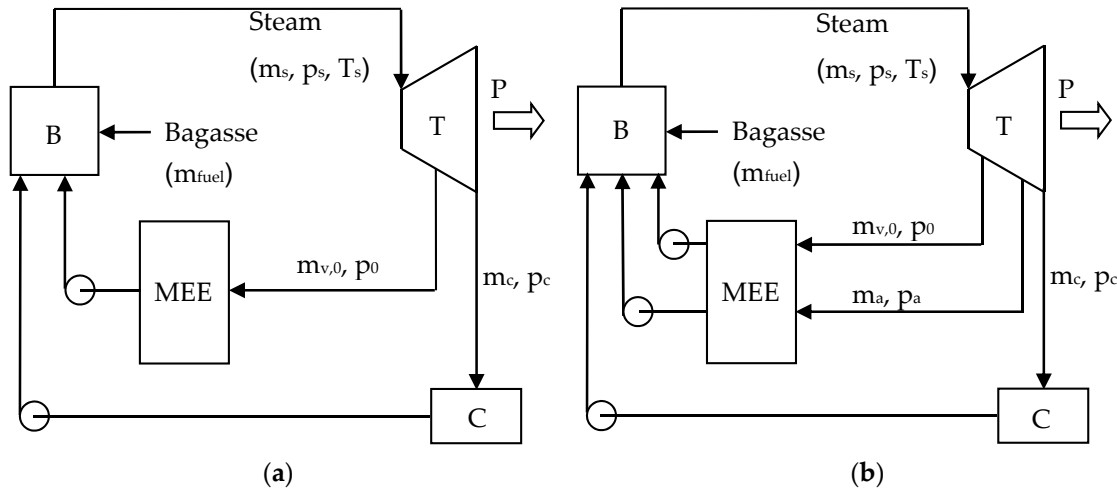

**Figure 3.** Cogeneration systems that use (**a**) the forward-feed multiple-effect evaporator (FF system) and (**b**) the backward-feed multiple-effect evaporator (BF system).

Both systems are required to process the same amount of sugar juice in order to produce the same amount of raw sugar. Furthermore, the juice concentrations at the inlet and outlet of forward-feed multiple-effect evaporator are the same as those of backward-feed multiple-effect evaporator, and the mass flow rates of evaporated water are the same for both evaporators. Under the conditions that the bagasse consumption rate ($m_{fuel}$), the inlet steam conditions ($p_s$ and $T_s$), and the condensing pressure ($p_c$) are the same in both systems, the energy efficiency parameter that may be used to compare both systems is power output. The power output of the FF system is

$$P = m_{v,0}(h_s - h_0) + m_c(h_s - h_c), \tag{51}$$

$$h_0 = h_s - \eta_t(h_s - h_{0s}), \tag{52}$$

$$h_c = h_s - \eta_t(h_s - h_{cs}), \tag{53}$$

where $\eta_t$ is isentropic efficiency of steam turbine, $h_s$ is specific enthalpy of steam at turbine inlet, $h_{0s}$ is specific enthalpy of the extracted steam at pressure $p_0$ and the same entropy as the inlet steam, $h_{cs}$ is specific enthalpy of the condensed steam at pressure $p_c$ and the same entropy as the inlet steam. The power output of the BF system is

$$P = m_{v,0}(h_s - h_0) + m_a(h_s - h_a) + m_c(h_s - h_c), \tag{54}$$

$$h_a = h_s - \eta_t(h_s - h_{as}), \tag{55}$$

where $h_{as}$ is specific enthalpy of the extracted steam at pressure $p_a$ and the same entropy as the inlet steam. Since $m_{fuel}$ is known, $m_s$ can be determined from boiler efficiency, which is defined as

$$\eta_b = \frac{m_s\left(h_s - h_{fw}\right)}{m_{fuel}HHV}, \tag{56}$$

where $h_{fw}$ is specific enthalpy of feed water, and *HHV* is the higher heating value of fuel.

## 5. Results and Discussion

The identical parameters of the forward-feed and backward-feed multiple-effect evaporators are $x_{in} = 15\%$, $x_{out} = 70\%$, $p_4 = 16$ kPa, and $T_{amb} = 30$ °C. The total evaporator surface area and the total juice heater surface area of the forward-feed multiple-effect evaporator are, respectively,

13,000 m² and 2500 m². In order to compare the performance of both systems, both evaporators should have the same total heating surface areas. Therefore, the total evaporator surface area of the backward-feed multiple-effect evaporator is 15,500 m². This requirement is based on the assumption that the investment cost of an evaporator depends mostly on its total heating surface area. The optimum distribution of total heating surface area that maximizes the mass flow rate of processed juice at a given exhaust steam pressure is to be determined for each evaporator.

Figure 4 shows the determination of the maximum mass flow rate of processed juice ($m_{f,max}$) in the forward-feed multiple-effect evaporator corresponding to the extracted steam pressure ($p_0$) of 200 kPa. Since the mass flow rate of processed juice ($m_{f,in}$) is a convex function of evaporator surface areas in the forward-feed multiple-effect evaporator, a line-search method can be used to find the optimum values of third-effect surface area ($A_3$) and second-effect surface area ($A_2$) that maximize the inlet juice mass flow rate ($m_{f,in}$) as shown in Figure 4a,b. Figure 4c shows that $m_{f,in}$ increases monotonically with decreasing first-effect surface area ($A_1$). However, vapor bled from the first effect must be supplied to the pan stage at a specified pressure ($p_1$). This constraint limits the maximum mass flow rate of processed juice ($m_{f,max}$) to 153.36 kg/s if $p_1$ is 150 kPa, as shown in Figure 4c. Figure 5 shows the determination of $m_{f,max}$ in the backward-feed multiple-effect evaporator corresponding to the extracted steam pressure ($p_0$) of 200 kPa. A line-search method can also be used to find the optimum values of $A_3$, $A_2$, and $A_1$ that maximize the inlet juice mass flow rate as shown in Figure 5a–c. It should be noted that there is no constraint in this case. It can be seen from Figure 5c that $m_{f,max}$ is 158.05 kg/s. Figure 6 shows variations of $m_{f,max}$ and $m_{v,0}$ with $p_0$ for the forward-feed multiple-effect evaporator that have the optimum distribution of heating surface areas and variations of $m_{f,max}$, $m_{v,0}$, and $m_a$ with $p_0$ for backward-feed multiple-effect evaporator that have the optimum distribution of heating surface areas. It can be seen that all mass flow rates decrease monotonically with $p_0$ for both evaporators. Furthermore, $m_{f,max}$ appears to be more sensitive to $p_0$ in the forward-feed multiple-effect evaporator than the backward-feed multiple-effect evaporator.

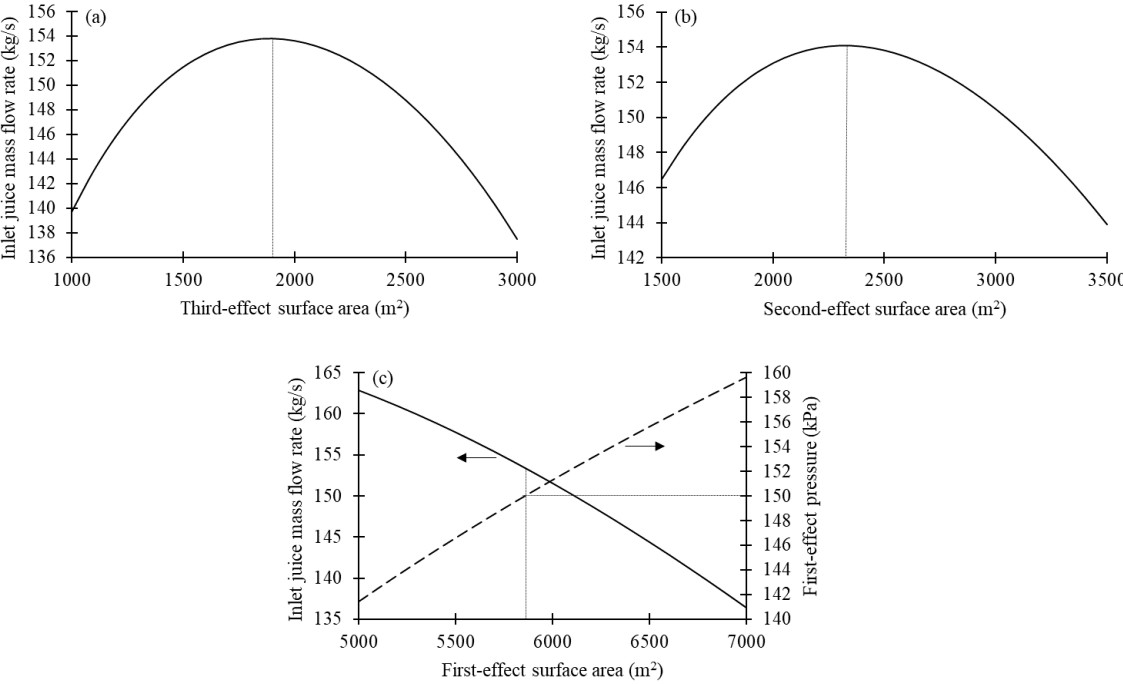

**Figure 4.** Procedure for finding the optimum surface area distribution in the forward-feed multiple-effect evaporator: (**a**) finding the optimum third-effect surface area ($A_3$), (**b**) finding the optimum second-effect surface area ($A_2$), and (**c**) finding the first-effect surface area ($A_1$) corresponding to the first-effect pressure ($p_1$) of 150 kPa.

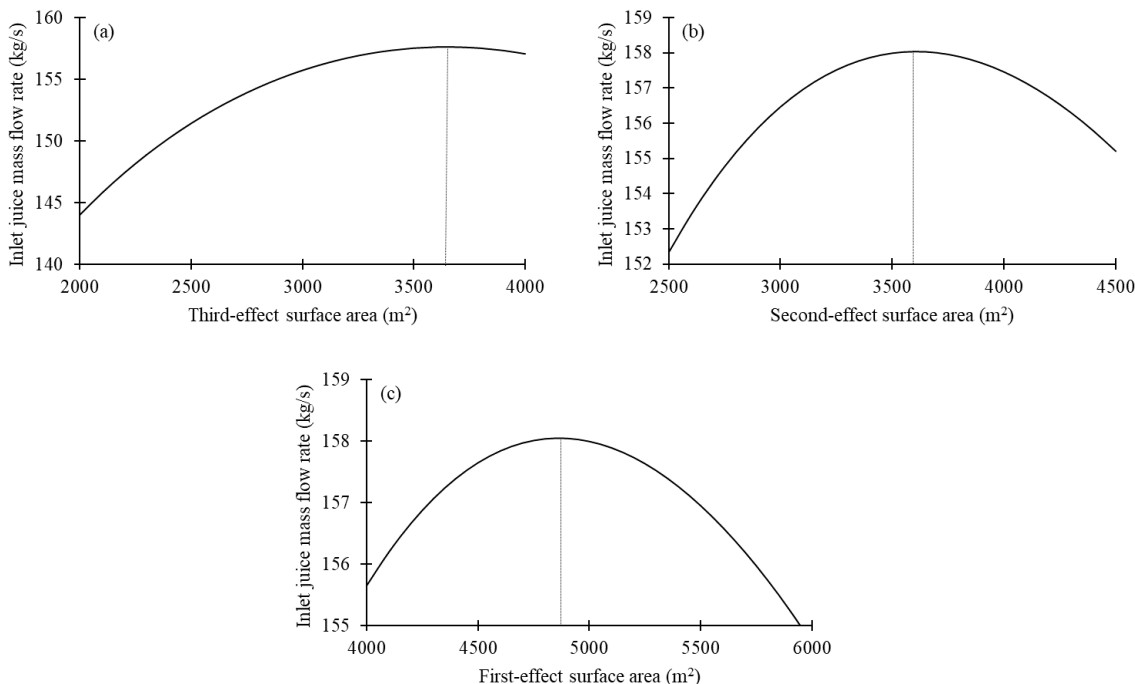

**Figure 5.** Procedure for finding the optimum surface area distribution in the backward-feed multiple-effect evaporator: (**a**) finding the optimum third-effect surface area ($A_3$), (**b**) finding the optimum second-effect surface area ($A_2$), and (**c**) finding the optimum first-effect surface area ($A_1$).

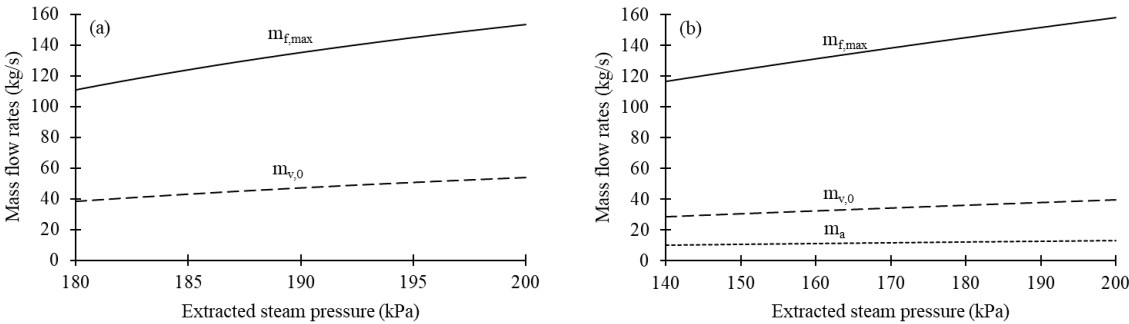

**Figure 6.** Variations of steam and juice mass flow rates with extracted steam pressure for (**a**) the forward-feed multiple-effect evaporator and (**b**) the backward-feed multiple-effect evaporator that have optimum distributions of heating surface areas.

The calculation of power outputs of the cogeneration systems requires information about fuel, steam conditions, boiler efficiency, and turbine efficiency. It is assumed that the mass flow rate of fuel consumed by both systems is 25 kg/s, the higher heating value of fuel is 9000 kJ/kg, the pressure and temperature of steam at boiler outlet are 4.5 MPa and 440 °C, the boiler efficiency is 70%, and the turbine efficiency is 85%. Figure 7 shows that the BF system produces more power output than the FF system that processes the same amount of sugar juice. Since both systems consume the same amount of fuel, the BF system is more energy efficient than the FF system. Table 1 compare the parameters of both systems when both evaporators have the optimum distributions of heating surface areas, and process 125 kg/s of inlet sugar juice. It can be seen that, under the conditions of identical total heating surface areas and juice processing capacities, the BF system requires less extracted steam than the FF system, and the evaporator pressures in the BF system are lower than those in the FF system. Furthermore, the power output of the BF system is 3.2% larger than that of the FF system. The increased power output is attributed to the backward-feed arrangement of the multiple-effect evaporator.

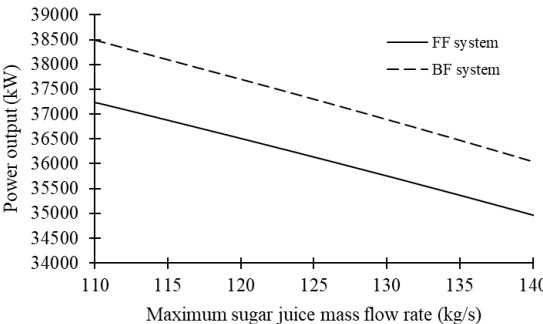

**Figure 7.** Variations of power outputs of the FF and BF systems with the maximum sugar juice flow rate.

**Table 1.** Simulation results at optimum operating conditions for the FF and BF systems that process 125 kg/s of inlet sugar juice.

|  | FF System | BF System |
|---|---|---|
| $A_1$ (m$^2$) | 7166 | 4884 |
| $A_2$ (m$^2$) | 1909 | 3597 |
| $A_3$ (m$^2$) | 1581 | 3455 |
| $A_4$ (m$^2$) | 2244 | 3564 |
| $A_{h,1}$ (m$^2$) | 405 | - |
| $A_{h,2}$ (m$^2$) | 2094 | - |
| $p_0$ (kPa) | 185.5 | 151.3 |
| $p_1$ (kPa) | 150.0 | 79.7 |
| $p_2$ (kPa) | 91.2 | 50.0 |
| $p_3$ (kPa) | 52.7 | 29.9 |
| $p_4$ (kPa) | 16.0 | 16.0 |
| $m_{v,0}$ (kg/s) | 43.45 | 30.62 |
| $m_a$ (kg/s) | 13.16 [1] | 10.63 [2] |
| $P$ (MW) | 36.14 | 37.31 |

[1] Vapor bled from the first effect at 150 kPa. [2] Extracted steam from turbine at 150 kPa.

Conventional cogeneration system in the sugar industry is the FF system. Even though it has been demonstrated in this paper that the BF system delivers more power output than the FF system under the same operating conditions, further analysis of costs and benefits of the BF system in comparison with the FF system is necessary before a decision can be made about the adoption of the BF system. Since both systems have the same total heating surface areas, the difference between the installation costs of both systems is due to the differences in the numbers of pumps and solution flash tanks. Figures 1–3 reveal that there are more pumps in the BF system than the FF system. Furthermore, the BF system requires three solution flash tanks, whereas none is required in the FF system. Therefore, installation cost of the BF system is expected to be slightly larger than that of the FF system. Whether the increased power output of the BF system compared with the FF system justifies the larger installation cost of the BF system compared with the FF system requires an economic analysis, which is beyond the scope of this paper. It should also be noted that steam and vapor pressures and temperatures in the backward-feed multiple-effect evaporator are lower than those in the forward-feed multiple-effect evaporator. Since sucrose inversion loss increases with temperature [1], there should be less sucrose inversion loss in the backward-feed multiple-effect evaporator. The reduction of sucrose inversion loss results in more production of raw sugar, which increases the revenue for the sugar factory, and provides additional justification for the installation cost of the BF system.

## 6. Conclusions

The main objective of this paper is to demonstrate that the backward-feed multiple-effect evaporator is more energy efficient than the conventional forward-feed multiple-effect evaporator that

has the same total heating surface area, and process the same amount of sugar juice. Mathematical models are developed for both evaporators for this investigation. Comparison is made between two cogeneration systems that use the forward-feed and backward-feed multiple-effect evaporators. Both evaporators have the optimum distributions of heating surface areas that yield the maximum inlet juice flow rates. Since each system consumes the same amount of fuel, and produces the same amount of raw sugar, the energy efficiency parameter is the power output. Simulation results indicate the forward-feed multiple-effect evaporator that has 13,000 m$^2$ of evaporator surface area and 2500 m$^2$ of juice heater surface area is capable of processing 125 kg/s of inlet juice flow rate using extracted steam at the pressure of 185.5 kPa and the mass flow rate of 43.45 kg/s. The cogeneration system that uses the forward-feed multiple-effect evaporator produces 36.14 MW of power output. The backward-feed multiple-effect evaporator that has 15,500 m$^2$ of evaporator surface area is capable of processing 125 kg/s of inlet juice flow rate using extracted steam at the pressure of 151.3 kPa and the mass flow rate of 30.62 kg/s. The cogeneration system that uses the backward-feed multiple-effect evaporator is also required to supply extracted steam at the pressure of 150 kPa and the mass flow rate of 10.63 kg/s for the pan stage. It produces 37.31 MW of power output. Therefore, the backward-feed multiple-effect evaporator is responsible for 3.2% more energy efficiency in this cogeneration system compared with the cogeneration system that uses the forward-feed multiple-effect evaporator.

**Funding:** This research received no external funding.

**Conflicts of Interest:** The authors declare no conflict of interest.

## Abbreviations

**Nomenclature**

| | |
|---|---|
| $A$ | heat transfer surface of evaporator, m$^2$ |
| $A_h$ | heat transfer surface of juice heater, m$^2$ |
| $c_p$ | specific heat capacity, kJ/kg.K |
| $H$ | juice level in evaporator, m |
| $HHV$ | juice level in evaporator, kJ/kg |
| $h$ | specific enthalpy, kJ/kg |
| $m$ | mass flow rate, kg/s |
| $P$ | power output, kW |
| $p$ | pressure, kPa |
| $T$ | temperature, °C |
| $U$ | heat transfer coefficient of evaporator, kW/m$^2$.K |
| $U_h$ | heat transfer coefficient of juice heater, kW/m$^2$.K |
| $x$ | concentration of sugar juice, % |

**Greek Symbols**

| | |
|---|---|
| $\varepsilon$ | heat loss coefficient in evaporator |
| $\eta_b$ | boiler efficiency |
| $\eta_t$ | turbine efficiency |
| $\rho$ | density, kg/m$^3$ |

**Subscripts**

| | |
|---|---|
| $a$ | vapor to pan stage |
| $b$ | vapor to juice heater |
| $c$ | vapor from flash tank, condenser |
| $e$ | flash evaporation |
| $f$ | sugar juice |
| $fw$ | feed water |
| $h$ | juice heater |
| $i$ | effect number |

| | |
|---|---|
| *l* | saturated liquid |
| *s* | Steam |
| *v* | saturated vapor |
| *vl* | vapor-to-liquid |
| *x* | juice heating inside evaporator vessels |

**Superscripts**

| | |
|---|---|
| *in* | inlet of an effect |
| *out* | outlet of an effect |

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
