# Peer review of "Increased Energy Efficiency of a Backward-Feed Multiple-Effect Evaporator Compared with a Forward-Feed Multiple-Effect Evaporator in the Cogeneration System of a Sugar Factory"

_processes, doi:10.3390/pr8030342_

Round 1
Reviewer 1 Report
This is a review of the manuscript “Increased Energy Efficiency of Backward-feed Multiple-effect Evaporator Compared with Forward-feed Multiple-effect Evaporator in Cogeneration System of Sugar Factory”.
This manuscript deals with a current topic that fits well in the scope of the journal. The subject may attract interest to the readers. In general, this manuscript is well organized and written, with comprehensive literature review, detailing the framework approach of the study, clearly stated methodology and nicely presented findings. The manuscript provides sufficient background information regarding the topic proposed.
However, the following requests/suggestions could be taken into account to improve the quality of the manuscript:
- it would be interesting your justified opinion regarding the practical applicability on a large scale of the proposed solution, namely "the cogeneration system that uses the backward-feed multiple-effect";
- you can highlight the usefulness of the research in the practical applicability;
- it would be useful for readers to critically analyze and discuss in detail the results obtained;
- also, the values presented in Table 1 should be provided with a discussion, comparative analysis that will highlight the performances related to the energy efficiency claimed by the author;
- check Figure 1... in the version of the manuscript available for review, it starts on page 2 - L92 and ends on page 3 - L107 (more than likely it is a word to pdf conversion error).
Having mentioned the above, this manuscript is proposed to be published after minor revision.
Author Response
Reviewer 1
This is a review of the manuscript “Increased Energy Efficiency of Backward-feed Multiple-effect Evaporator Compared with Forward-feed Multiple-effect Evaporator in Cogeneration System of Sugar Factory”.
This manuscript deals with a current topic that fits well in the scope of the journal. The subject may attract interest to the readers. In general, this manuscript is well organized and written, with comprehensive literature review, detailing the framework approach of the study, clearly stated methodology and nicely presented findings. The manuscript provides sufficient background information regarding the topic proposed.
However, the following requests/suggestions could be taken into account to improve the quality of the manuscript:
- it would be interesting your justified opinion regarding the practical applicability on a large scale of the proposed solution, namely "the cogeneration system that uses the backward-feed multiple-effect";
- you can highlight the usefulness of the research in the practical applicability;
- it would be useful for readers to critically analyze and discuss in detail the results obtained;
Response
A paragraph providing opinion about the practical applicability of the proposed process modification is added at the end of Section 4. Critical discussion of the results is also contained in this paragraph.
- also, the values presented in Table 1 should be provided with a discussion, comparative analysis that will highlight the performances related to the energy efficiency claimed by the author;
Response
For comparative analysis, evaporator effect pressures of both evaporators are added to Table 1 Additional discussion of Table 1 is also provided in the revised manuscript.
- check Figure 1... in the version of the manuscript available for review, it starts on page 2 - L92 and ends on page 3 - L107 (more than likely it is a word to pdf conversion error).
Response
It is indeed the file conversion error.
Reviewer 2 Report
The paper is well written with clear math derivations and simulation results. All the results and background are demoed clearly in appropriate way. I enjoy reading it. Only some suggestions are as the follows:
- Figure 1 is located on two different pages. Please re-organize it.
- Figures 4-6 has very long titles with explanation of operating and initial conditions of results. I suggest those contents can be inserted in the main body or figures themselves.
- In the conclusion, the author state that the backward-feed multiple-effect evaporator is responsible for 2.5% more energy efficiency than forward-feed one. 2.5% is not a very big energy efficiency improvement in the practical application. Also in practice, the industry has to consider the cost difference of establishing and operating the backward-feed and forward-feed systems. If there is particular high cost on backward-feed system, we may probably see the value of energy efficiency improvement and return. I suggest that the author need add some cost analysis of both systems.
Author Response
Reviewer 2
The paper is well written with clear math derivations and simulation results. All the results and background are demoed clearly in appropriate way. I enjoy reading it. Only some suggestions are as the follows:
- Figure 1 is located on two different pages. Please re-organize it.
Response
It is the file conversion error.
- Figures 4-6 has very long titles with explanation of operating and initial conditions of results. I suggest those contents can be inserted in the main body or figures themselves.
Response
Captions of these figures are shortened, and additional explanations of them are added in the revised manuscript.
- In the conclusion, the author state that the backward-feed multiple-effect evaporator is responsible for 2.5% more energy efficiency than forward-feed one. 2.5% is not a very big energy efficiency improvement in the practical application. Also in practice, the industry has to consider the cost difference of establishing and operating the backward-feed and forward-feed systems. If there is particular high cost on backward-feed system, we may probably see the value of energy efficiency improvement and return. I suggest that the author need add some cost analysis of both systems.
Response
A paragraph providing cost analysis of both systems is added at the end of Section 4.